# Characterizing Ambient Seismic Noise in an Urban Park Environment

**DOI:** 10.3390/s23052446

**Published:** 2023-02-22

**Authors:** Benjamin Saadia, Georgia Fotopoulos

**Affiliations:** Department of Geological Sciences and Geological Engineering, Queen’s University, Kingston, ON K7L 3N6, Canada

**Keywords:** urban characterization, ambient seismic noise, geophysical surveys, wavelets

## Abstract

In this study, a method for characterizing ambient seismic noise in an urban park using a pair of Tromino3G+ seismographs simultaneously recording high-gain velocity along two axes (north-south and east-west) is presented. The motivation for this study is to provide design parameters for seismic surveys conducted at a site prior to the installation of long-term permanent seismographs. Ambient seismic noise refers to the coherent component of the measured signal that comes from uncontrolled, or passive sources (natural and anthropogenic). Applications of interest include geotechnical studies, modeling the seismic response of infrastructure, surface monitoring, noise mitigation, and urban activity monitoring, which may exploit the use of well-distributed seismograph stations within an area of interest, recording on a days-to-years scale. An ideal well-distributed array of seismographs may not be feasible for all sites and therefore, it is important to identify means for characterizing the ambient seismic noise in urban environments and limitations imposed with a reduced spatial distribution of stations, herein two stations. The developed workflow involves a continuous wavelet transform, peak detection, and event characterization. Events are classified by amplitude, frequency, occurrence time, source azimuth relative to the seismograph, duration, and bandwidth. Depending on the applications, results can guide seismograph selection (sampling frequency and sensitivity) and seismograph placement within the area of interest.

## 1. Introduction

Active seismic surveys measure the signal from a source with a predetermined location and frequency to probe the surface and subsurface [1]. Passive seismic surveys measure ambient seismic noise, which includes a random component (not spatially or temporally correlated across stations) and a coherent component [2]. Ambient seismic noise refers to the coherent component of the measured signal that comes from uncontrolled, or passive sources such as natural (e.g., wind, water waves, tides) and anthropogenic sources (e.g., traffic, pedestrians, machinery) [2,3]. Ambient seismic noise includes a broad range of frequencies and amplitudes depending on the proximity of the noise source in relation to the seismograph stations, the source mechanism, and the path and materials between the source and the receiver (see Figure 1). The seismic source mechanism produces signals of varying frequencies and amplitudes at a given azimuth to the receiver. As the signal travels along a path to the receiver, it attenuates and disperses (depending on subsurface materials). The receiver records the signal with a given sensitivity, sampling frequency, and detection range, resulting in the seismic recording.

In order to achieve the overall goal of characterizing the ambient seismic noise in an urban park environment, a new methodology for characterizing events by amplitude, frequency, time, source azimuth relative to the seismograph, duration, and bandwidth was developed. Conventional techniques for seismic event detection and characterization include waveform correlation which involves measuring the similarity between a template seismic event and a target seismic acquisition [4,5,6,7], and amplitude ratio techniques which involve computing the ratio of the short-term time average to that of a long time average of signal amplitude and identifying an event when the ratio exceeds a predetermined threshold [8,9,10,11]. For the purposes of this paper, an event is a coherent signal used to estimate amplitude and source azimuth. It is also assumed that the influence of the various subsurface materials on the signal characteristics is negligible. A catalog of events is created in an urban park environment using a two-seismograph setup. Applications include informing the design of a long-term ambient noise monitoring system by providing information about the dominant frequencies, amplitudes, and seismic source locations in and around the site. In this case, an urban park provides the ideal site for a case study given the variety of usage zones on its periphery. However, the work presented in this study is not park-specific and is intended for general application at any site. Examples of applications for ambient seismic noise monitoring include geotechnical studies [12,13,14,15,16,17,18,19], near-surface imaging using traffic-induced surface waves [20,21], infrastructure monitoring [22,23,24], surface monitoring [25,26,27,28,29], noise mitigation [30,31,32], and urban activity monitoring [33,34,35,36,37]. While long-term, multi-day acquisitions are ideal, this study aims to assess short-term (hours-scale) acquisitions to inform future seismograph installations. By improving the design of long-term seismograph installations through a-priori data collection aimed at identifying the presence of events, there is the potential to collect higher quality data for all aforementioned applications. In the following section, a description of the field data collection campaign for this study is provided.

## 2. Materials and Methods

### 2.1. Description of Field Data

Two Tromino3G+ seismographs (herein referred to as stations T1 and T2) simultaneously recorded high-gain velocity (mm/s) along two axes (north-south and east-west) with a sampling frequency of 512 Hz. This sampling frequency was deemed sufficient for the area of interest; however, the methodology applies to any sampling frequency. The manufacturer technical specifications for the seismographs are provided in Table 1 [38].

The seismographs were coupled to the ground through metal spikes in an urban park (City Park in Kingston, Ontario, Canada) spanning an area of approximately 100,000 m^2^ (370 m × 270 m) surrounded by two-way vehicle roadways and urban activity such as pedestrians and cyclists. See Figure 2 for a map of City Park, including the locations of the seismograph stations T1 and T2, traffic flows, and usage zones classified according to the primary type of activity. The zones include a hospital complex (industrial generators and emergency vehicle traffic), Lake Ontario (boats and waves), recreational areas (pedestrians, cyclists, reduced vehicle traffic, playground), residential (light vehicles), and a university district (pedestrians, timed events such as increased pedestrian traffic during regular class transitions, generators). 

Throughout the duration of the recordings, field notes about vehicle traffic, pedestrian activity, aircraft, and wind were collected and timestamped. The field notes were later correlated to events in the seismograph recordings to improve the interpretation of the results. Prior to data acquisition, the seismographs were calibrated and tested, and two data sets of differing acquisition times were collected. The seismograph stations were fixed at two reference points 20 m apart in the center of the park. The 20 m distance between stations was chosen based on a number of experimental surveys and observations in the park and was deemed to be sufficient. Two seismographs recorded simultaneously at each station increasing the probability of capturing seismic events from weak seismic sources (such as pedestrian activity) that may otherwise not exhibit an amplitude above the instrument noise floor due to the relative position of the seismograph and the source. Ambient seismic noise (i.e., no active sources present) was recorded over a span of three hours, from 15:00 to 18:00 EST/local time on 16 November 2021. A second data collection campaign occurred with the seismograph stations recording for 20 min at six different locations within the park bounds. These locations ranged from 80 to 170 m radially in all directions from the reference point, with 20 m between seismographs located at the same station. This data was collected from 15:00 to 18:00 EST on 21 November 2021. At the site, variations in the intensity of activity are present depending on time of day (e.g., at night when vehicle and pedestrian traffic are reduced relative to the afternoon). The same time interval was chosen to minimize possible differences due to daily and weekly variations in ambient noise and to ensure representative results of a high-activity period.

During the acquisitions, detailed field observations were made regarding the origins of audible and visible activities in the park. With the exception of a children’s ball game, there was no anomalous activity within visible range (50 m) of the seismograph stations. Frequent and consistent traffic was visually observed running EW to the south of the park, while traffic to the north of the park was visually observed to be dominated by slower-moving vehicles passing less frequently. Within the park, pedestrian activity was concentrated along the EW path in the middle of the site based on a manual count of pedestrians passing throughout the acquisition period. The acquisition time ranges for typical applications aggregated from 25 different published case studies are plotted in Figure 3. Applications with a time range that exceed this study’s three-hour acquisition time (highlighted in solid blue in Figure 3), such as noise studies and urban activity monitoring, will benefit the most from the methodology presented in this study because this study’s acquisition time is a small fraction of the total target application’s acquisition time. 

In total, 14 stations collected 28 recordings (NS and EW axes). Table 2 provides each seismograph’s ID, start and end time (EST), duration (minutes), and location (°, m) with reference to E1A T1, the reference station.

Figure 4 provides a visual overview of the recordings. Each 20 min recording (panels e, f, g, and h) contains 614,399 measurements, and each 180 min recording (panels a, b, c, and d) contains 5,529,599 measurements. Each column in Figure 4 refers to the seismograph station (a, c, e, and g for T1 and b, d, f, and h for T2) with locations in the park given in Figure 2, and the blue and yellow time series correspond to the measurement direction (NS and EW, respectively). The horizontal axes refer to local time of day between 15:00 and 18:00 on 16 November 2021 (E1A) or 21 November 2021 (E1Bx), and the vertical axes refer to velocity (mm/s). It should be noted that parts of the time series without data represent the time between moves.

In the next section, the processing method for extracting events from the seismic recordings is provided. Note that while amplitude spikes are evident in the seismic recordings, the method presented in the next section decomposes the recordings into their constituent frequencies using the continuous wavelet transform and identifies events classified by occurrence time, frequency, bandwidth, duration, and azimuth. This allows for an analysis of the dominant frequencies and azimuths throughout the site. Extracting events from the time-frequency spectrum allows for a more detailed approach to characterizing urban ambient seismic noise, identifies major ambient noise sources, and allows bounds on instrument sensitivity and bandwidth to be set.

### 2.2. Processing Methodology for Event Identification

Figure 5 provides a graphical overview of the developed processing methodology used to identify time-frequency dependent events from a recording of ambient seismic noise. The seismic records were detrended via a least-squares adjustment and residuals were analyzed. The methodology can be split into three sections: wavelet transform, peak detection, and event characterization. Equations used are shown in the following section.

#### 2.2.1. Continuous Wavelet Transform

Let $\vec{w}$ be the set of detrended velocity records for a given seismograph, given by:(1)$$\vec{w}=\left[\begin{array}{ccc} {\vec{v}}_{NS} & {\vec{v}}_{EW} & {\vec{v}}_{Z} \end{array}\right]$$
where $\vec{v}$ is the time series of measured velocity values (mm/s) in the North-South, East-West, and vertical directions. The continuous wavelet transform is used to obtain the time-frequency spectrum of every velocity record $\vec{v}$ which is common for the analysis of ambient seismic noise [39,40,41]. The absolute values of the wavelet coefficients from each wavelet transform are used to obtain scalogram amplitudes of events (mm/s), differentiated by signal frequency (Hz). Due to spectral leakage, the scalogram amplitudes are not identical to the measured velocities; however, for the purposes of this paper, amplitude and scalogram amplitude obtained by taking the absolute value of the wavelet coefficients are used interchangeably. The amplitudes are used in conjunction with a user-defined threshold value to define the location in time-frequency space of an event (see Section 2.2.2). Due to its direct relationship between scale and center frequency, the complex Morlet wavelet ψ [41] was chosen to be the mother wavelet in the analysis as follows:(2)$$\psi \left(t\right)=\frac{1}{\sqrt{\pi f_{b}}}\exp\left(2\pi if_{c}t\right)\exp\left(\frac{-x^{2}}{f_{b}}\right)$$
where $f_{b}$ is the time-decay parameter and $f_{c}$ is the center frequency (Hz) of the wavelet. The wavelet scales are given by Equation (3) as provided in [41] and shown below:(3)$$s_{m}=s_{0}2^{m\delta m}\;\;\;\;\;\;\;\;\;\;\;\;\;\;\;\;\;\;\;\;\;\;\;\;m=0,\;\ldots ,\;M$$
where $s_{0}$ is the minimum scale, δm is the scale resolution, and M is the number of scales. The wavelet coefficients are computed in [41] via:(4)$$F\left(\tau ,s_{m}\right)=\frac{1}{\sqrt{\left|s_{m}\right|}}\overset{+\infty }{\underset{-\infty }{\int }}\vec{v}\psi *\left(\frac{t-\tau }{s_{m}}\right)dt$$
where $\vec{v}$ and ψ are convolved at each scale $s_{m}$ and τ represents the position of the wavelet in the time domain. This provides the wavelet coefficients, which are converted to amplitude by taking the absolute value of the real-valued components of the coefficients. Let the scalogram amplitudes for each record be given:(5)$$\vec{a}=\left[\begin{array}{ccc} A_{NS} & A_{EW} & A_{Z} \end{array}\right]$$
where A is the m by k matrix of wavelet coefficients for each scale, $s_{m}$. Each scale is related to the wavelet center frequency as follows:(6)$$f_{m}=\frac{f_{c}}{s_{m}}$$
where $f_{m}$ is the signal frequency (herein simplified to f) and $f_{c}$ is the center frequency of the wavelet. With the scalogram amplitudes obtained for every velocity record, the peak-finding algorithm is initiated (see Section 2.2.2).

#### 2.2.2. Peak Detection

The purpose of this processing step is to determine the local maxima in terms of amplitude within time-frequency space. First, a 2-dimensional maximum filter in Equation (7) is applied to the scalogram amplitude values:(7)$$A^{\prime }=\max\left\{A\left(t+u,\;f+v\right)\right\}$$
where $A^{\prime }$ is the maximum filtered scalogram amplitude matrix, and u and v are user-defined distances defining the neighbourhood size for computing the maxima in the time and frequency domains, respectively. $A^{\prime }$ is compared to A to identify the coordinates of the peaks as follows:(8)$$\left(A^{\prime }\left(t,f\right)=A\left(t,f\right)\right)\wedge (A^{\prime }\left(t,f\right)>a_{max})\to \vec{e}\left(a,t,f\right)$$
where $a_{max}$ is a user-defined amplitude threshold and $\vec{e}$ is an event defined by its amplitude a, time t, and frequency f. Let the events for each record in $\vec{w}$ be given by:(9)$$\vec{c}=\left[\begin{array}{ccc} {\vec{e}}_{NS} & {\vec{e}}_{EW} & {\vec{e}}_{Z} \end{array}\right]$$
where $\vec{c}$ is the set of events $\vec{e}$ in each measurement direction for each $\vec{w}$. Given each event defined in amplitude, time, and frequency, event characterization is conducted as described in the following section.

#### 2.2.3. Event Characterization

Each event is characterized according to its frequency bandwidth Δf, duration Δt, and azimuth θ with respect to North. To obtain the duration of an event, local peaks along the time axis in A where $\vec{e}$ is present are calculated. Note that this is a separate peak detection step from the 2D peak detection used in Section 2.2.2 to identify the location of the events in time-frequency space. In the 1D peak detection step in this section, it is conducted along the time axis of each event identified in Section 2.2.2. Let the 1D function representing the amplitudes at a certain frequency be given by $A_{f}\left(t\right)$, where f is the frequency and t is the time. Note that there is a new $A_{f}\left(t\right)$ for every $\vec{e}$. Let the set of peak amplitudes along the time axis be given as:(10)$$\vec{q}=\left[\begin{array}{cccc} q_{i} & q_{i+1} & \ldots & q_{n} \end{array}\right]\;\;\left(i=1,\;\ldots ,\;n\right)$$
where $q_{i}$ is the amplitude of the peak and n is the number of peaks along the time axis. The 1D peak detection in this calculation step is performed using the same $a_{max}$ as in Section 2.2.2, but u or v are set to a minimum rather than the user-defined values. To calculate the prominence of the peak, two intervals must be defined by the following condition:(11)$$\begin{array}{c} k_{R}\\ k_{L} \end{array}=\begin{array}{c} \left(q_{i}<q_{i+1})\vee \left(i+1=n\right)\to [q_{i},q_{i+1}\right]\\ \left(q_{i}<q_{i-1})\vee \left(i-1=1\right)\;\to [q_{i},q_{i-1}\right] \end{array}\;\;\;\;\;\;\;\;\;\;\;\;\;\;\left(i=1,\;\ldots ,\;n\right)$$
where $k_{R}$ is the interval to the right of the peak and $k_{L}$ is the interval to the left of the peak. Next, prominence is calculated by:(12)$$p=q_{i}-\max\left\{\begin{array}{c} \min[a_{i},a_{i+1}]\\ \min[a_{i},a_{i-1}] \end{array}\right\}\;\;\;\;\;\;\left(i=1,\;\ldots ,\;n\right)$$
and the amplitude at which to evaluate the width measurement is found by:(13)$$a_{eval}=a-p*0.5$$
where $a_{eval}$ is the aforementioned evaluation amplitude and a is amplitude of the event in $\vec{e}$. The points of intersection between $a_{eval}$ are:(14)$$a_{eval}=A_{f}\left(t_{a_{eval}}\right)\to t_{l},t_{r}$$
where $t_{l}$ and $t_{r}$ are the two intersection points to the left and right of $q_{i}$, respectively. Finally, the duration of the event is calculated by:(15)$$\Delta t=t_{r}-t_{l}$$

To calculate Δf, the same process is performed, except that the local peaks are calculated along the frequency axis (time held constant). The azimuth of an event on the NS and EW axes are calculated with Equations (16) and (17), respectively:(16)$$\theta_{{\vec{e}}_{NS}}=\tan^{-1}\left(\frac{a_{EW}^{\prime }}{a_{NS}}\right)$$
(17)$$\theta_{{\vec{e}}_{EW}}=90{}^{\circ}-\tan^{-1}\left(\frac{a_{NS}^{\prime }}{a_{EW}}\right)$$
where $a_{EW}^{\prime }$ is the amplitude sampled from $A_{EW}$ at the f and t of the event. Note that the azimuth values are relative to the North-South axis but can be in any quadrant.

## 3. Results

Summary statistics for each seismic recording are presented in Figure 6 and are analyzed prior to event characterization in order to assess the seismograph positioning and the spatial scale of the analysis. The distribution of events based on defined characteristics is examined, comparing frequency to amplitude, bandwidth, and azimuth. The experimental parameters used for all calculations are provided in Table 3. The selected parameters were determined to produce a number of events for each seismograph recording. The amplitude threshold was selected to be greater than the instrument noise floor of 0.023 mm/s.

Along the EW measurement axis, root mean square (RMS) velocity is greatest for E1B4 T2 (0.0066 mm/s) and is closely followed by E1B5 T1 (0.0057 mm/s) and E1B2 T1 (0.0056 mm/s) (Figure 6a). For both NS and EW directions, RMS values averaged across seismographs are 1.5–2 times higher for E1B2, E1B4, and E1B5. Given that RMS is proportional to energy, this is consistent with both the proximity to streets and acquisition time relative to peak traffic time. These values are within the same order of magnitude as other ambient noise studies conducted in urban environments (see [42,43]). Between seismographs at the same station (in the extreme case, E1B5), the RMS varies by a maximum of 58% and 45% along the NS and EW axes, respectively, with reference to the maximum RMS. In this case, the placement of the seismographs (20 m apart) impacts the total measured energy. However, for most of the stations, the impact of seismograph placement did not significantly change the RMS, implying that the choice to analyze a particular seismograph’s measurements at a given station is not likely to cause variability in the results.

The standard deviations (STD) of the recordings at each station between seismographs are within 38% along the NS and EW axes with reference to the maximum STD, with the exception of E1A T1 in the EW direction, which is 82% (Figure 6b). The anomalously large STD for E1A T1 EW may be due to its proximity to a busy pedestrian path from which observed pedestrian activity caused several amplitude deviations from the mean. The STD values demonstrate that the observed velocities are generally within the range of 0.015 mm/s from the mean. Higher STD values indicate that the observations include a variety of amplitudes, which likely correlate to more varied activities at the site. Consistent STDs across the stations in the analysis imply that combining the results from the analysis in Section 3 does not over-represent a particular spatial domain of the site. If, for example, stations positioned in the north of the site exhibited significantly larger STDs, this would make the case for dividing the analysis into two distinct zones.

Peak amplitudes at each station between seismographs vary by a maximum (E1B2) of 80% and 63% along the NS and EW axes, respectively, with reference to the maximum peak amplitude (Figure 6c). The considerable variation in peak amplitudes between T1 and T2 at each station may be due to proximity to the event source and the attenuation of the signal in the subsurface. In general, the RMS results are more consistent across seismographs at the same station, highlighting that RMS is a more robust metric for assessing seismograph reliability than peak amplitude, which overemphasizes specific events. However, RMS is a metric that summarizes the entire recording, meaning that the impact of an individual event on the RMS may be reduced. For this reason, discussion of the RMS is limited to the reliability of results between the seismographs at each station. Anomalous events with high amplitudes may be important to the application of the study and therefore, are not removed in the preliminary characterization. However, note that distribution figures discussed in this section prioritize displaying the majority of events, rather than displaying every event. The noise floor for the seismographs was calculated to be 0.023 mm/s based on the sensitivity and dynamic range of the seismographs. The amplitude threshold parameter must be set above the noise floor and is selected to include low-amplitude events in this study. The purpose of this is to characterize all events to understand broad trends in the ambient noise rather than to detect anomalous events. The following sections consider the distributions of events obtained by applying the processing methodology outlined in Section 2.2.

### 3.1. Frequency vs. Amplitude

Figure 7 provides the frequency-amplitude distribution (red contours representing steps of 10% probability density) in Hz and mm/s for the events in all recordings. There are two clusters of events at 30 Hz and at 48 Hz, and 95% of events occur with a frequency of less than 45 Hz. The clusters of events at different frequencies may be due to the presence of two distinct source mechanisms around the site. Given that the data was recorded with a Nyquist frequency of 256 Hz, the 45 Hz 95% cutoff indicates that the high-frequency signals attenuate to an amplitude below 0.024 mm/s prior to reaching the seismographs. The highest amplitude events occur between 25–35 Hz with 85% of events occurring with an amplitude of less than 0.05 mm/s. This is within the frequency range expected for traffic-induced surface waves, which dominate in this urban environment. The presence of lower-amplitude events below 15 Hz is consistent with pedestrian activity on the paths near the receivers and the observation that that the timing and magnitude of amplitude change can be different for receivers positioned 20 m apart.

Figure 8 divides the frequency-amplitude distributions by measurement direction. The NS events (Figure 8a) exhibit a broader distribution of frequencies than the EW events (Figure 8b). Additionally, the majority of NS events are clustered between 15–25 Hz and 40–50 Hz, while the EW events are grouped in a single cluster between 25–35 Hz. Vehicular traffic is likely the dominant source given that it is documented to produce signals within this frequency range [33,34,37]. For the EW events, amplitudes above 0.05 mm/s are concentrated within this frequency range, while the NS measurements have fewer high-amplitude events. This is found to be due to the EW-parallel roads which were observed to have more vehicle traffic than the NS-parallel roads.

Examination of the frequency-amplitude distributions at individual stations reveals the dependence of detectable frequencies on station location within the site. For example, the northern edge of the park (Figure 9a) exhibits two frequency clusters, ranging from 5–15 Hz and 25–35 Hz. This may be due to the EW-running road to the north of the station, which was observed to have light vehicular traffic and heavy pedestrian activity. In the SE corner of the park (Figure 9b) one cluster at 30 Hz demonstrates higher amplitudes, likely due to frequent vehicular traffic moving at consistent speeds and less observed pedestrian activity. Events ranging from 5–15 Hz indicate the presence of increased pedestrian activity in this sector, which is documented to produce signals in this frequency range [44,45]. This is consistent with the locations of the seismographs relative to the busiest footpaths that run along the EW axis and observed pedestrian activity during the recordings.

### 3.2. Frequency vs. Bandwidth

Figure 10 provides the frequency-bandwidth distribution of events for the NS (a) and EW (b) seismic recordings. It is observed that the bandwidth of events increases at higher frequencies, with two clusters at different bandwidths between 15–25 Hz for NS and 25–35 Hz for EW. This is evidence for the frequency clusters being the result of separate sources because it implies that their source mechanisms generate a different range of signal frequencies.

### 3.3. Event Azimuth

The results for the frequency-azimuth distribution of events are given in Figure 11a,b providing the possible source directions overlaid on the map of the site. The azimuth represented herein is the angle of the event in any quadrant relative to the NS axis of the seismograph and provides an indication of whether an event is located closer to parallel with the NS or EW axis. The azimuth is calculated using the NS and EW channels of the seismograph recording the event. The reason for limiting the analysis to one seismograph at a time is due to the scalogram amplitudes excluding information about the polarity of the detected signals. There are three clusters of events occurring at different azimuths and frequencies. Most events occur at 80° from the NS axis with a frequency between 25–35 Hz, while at 20°, the events occur between 40–50 Hz. There is also a cluster at 75° and 5–15 Hz. The presence of three distinct frequency-azimuth clusters is evidence of multiple dominant ambient noise sources with a distinct spatial distribution at the site.

Examining the azimuth-amplitude distribution in Figure 12a, there is further evidence that the ambient noise anisotropy is the result of multiple sources rather than subsurface differences. Of the events with an amplitude greater than 0.05 mm/s, 50% occurred at azimuth angles steeper than 77°. Additionally, Figure 12b shows the azimuth-duration distribution of all events. Of the events with a duration longer than 1 s, 50% occurred at angles steeper than 76°.

The higher amplitudes and longer duration of events at steeper angles provide additional evidence for multiple dominant ambient noise sources with a distinct spatial distribution at the site. If there is a wide azimuth distribution that is localized by frequency (as in Figure 13a), this indicates that the signals of different frequencies originate in different locations, probably from different sources. In the case of a, the 5–15 Hz events at 15–30° likely arise from the pedestrian footpaths around the seismographs. The vehicular traffic on the road running EW to the south of the seismograph is the probable cause of the 25–35 Hz events at the same angles. There is also a cluster of 15–35 events at 60–80°, which originate from vehicle traffic running NS to the east of the seismographs. Figure 13b also shows a broad distribution of azimuths with one cluster of frequencies at 15–25 Hz, but there is no clear relationship between frequency and azimuth. This is likely due to both the NS and EW roads having a similar type of vehicular traffic, which acts as the ambient noise source.

## 4. Discussion

Based on distributions of parameters for the detected events, there are several findings that inform the design of long-term or permanent ambient noise monitoring system at the site. Firstly, bounds on instrument sampling frequency and sensitivity can be established. Given that 95% of events occur with a frequency below 45 Hz with a minimum frequency of 4 Hz, there is no need to use a sampling frequency higher than 90 Hz or lower than 8 Hz. Because 85% of events occur with a velocity of less than 0.05 mm/s, a permanent ambient noise monitoring system must be capable of detecting low amplitude vibrations. Its sensitivity (ratio between physical input to electrical output) need not be designed for large amplitude shocks. Given that velocity amplitude is proportional to energy, this is also potentially and directly useful for modeling and monitoring infrastructure health.

Secondly, three distinct noise sources can be identified and characterized. There are two dominant frequencies (in terms of amplitudes) at the site, ranging from 15–35 Hz and 35–45 Hz, with a majority of events occurring in the former. The absence of frequency clusters above 45 Hz implies that industrial equipment such as generators or construction machines are not significant sources at the site [33,46]. There is also a prominent cluster at 5–15 Hz, which is lower amplitude and is present in recordings close to pedestrian footpaths. There is a broad spread of frequencies across different azimuths to the seismograph, but there is a trend that steeper azimuths from the NS axis correlate to the 5–15 Hz and the 15–35 Hz frequency range. Events detected in the EW recordings show more events and higher amplitudes at lower frequencies, which is consistent with the azimuth distribution. Events longer than 1 s and with amplitudes above 0.05 mm/s tend to occur at steep angles. These results, combined with the field observations recorded at the site, demonstrate that there are three distinct types of ambient noise source at the site, summarized in Table 4. 

Based on the clear frequency localization of the events at some stations, the 15–35 Hz and 35–45 Hz noise sources are predominantly vehicular traffic, with different amplitudes and frequencies observed along residential access (NS-running) and transportation (EW-running) roads. The 5–15 Hz cluster is likely due to pedestrian activity. Stations in the north and SE of the park exhibit different dominant frequencies (Figure 9) and display different degrees of frequency localization in space (Figure 13. The events of 15–35 Hz originate at steep azimuths, likely to the east of the seismograph where there is a residential road. The events of 35–45 Hz occur at more shallow angles, likely from the road to the south of the seismograph, which has more vehicular traffic. It is clear that the location of the seismograph station relative to the source has an impact on the detected events. If the ambient noise sources were constant around the site, the location of the stations would not be expected to drastically change event detection. Additionally, different event bandwidths for the dominant frequencies also provide evidence for the multiple-source scenario. The bandwidth likely relates to the source mechanism because it will generate a different spread of signal frequencies depending on the type of source. If the event bandwidth were primarily dependent on subsurface materials, the observed bandwidths would not correlate with the dominant frequencies.

Understanding the ambient noise source spatial distributions has applications for permanent ambient noise monitoring systems. If the goal is to monitor the source activity (for example, traffic monitoring), then the seismographs should be positioned close to the streets to differentiate the sources by increasing the signal-to-noise ratio [37,47]. If the goal is to monitor the origin of the strongest ambient noise source in the park, the seismograph should be located in the center of the park. When precise source localization is important for the application, more than two seismographs are required to triangulate the absolute source azimuth. However, this study demonstrates that two seismographs recording simultaneously in varying locations can produce estimates of source azimuth when combined with field observations that are useful to understanding the general distribution of events. For the application of designing a permanent ambient noise monitoring system, a uniform spatial distribution of receivers throughout the park will detect more events along its EW axis. Thus, the event detection amplitude threshold must be set based on the minimum amplitude of the desired event along the NS axis. Additionally, to detect more NS events, an ambient noise monitoring system is recommended to have more seismographs positioned in the North and South of the park.

## 5. Conclusions

A methodology for characterizing ambient seismic events was developed. The work-flow employs a peak detection algorithm applied to the continuous wavelet transform-derived amplitudes to identify the events in time and frequency space. The source azimuth relative to the seismograph, duration, and bandwidth were calculated for each event. By analyzing the distribution of events according to each parameter, characteristics about the site were observed. The application of the developed methodology to the field data collected in City Park, Kingston, Ontario with a complex ambient seismic noise profile (due to a combination of anthropogenic sources) resulted in consistent results at different stations in the NS and EW measurement directions. The overall goal of cataloging events in an urban park environment according to their event parameters was accomplished and used to provide recommendations for the design of a permanent ambient noise monitoring system. A wide variety of sensors are available for these surveys, each with different bandwidths (10s of mHz to kHz) and ranges (for example, micro-electromechanical sensors tend to be better at detecting strong motions above 9.8 m/s^2^). The recommendations included selecting seismographs with a bandwidth capable of measuring signals up to 45 Hz and a sensitivity and range capable of measuring amplitudes <0.05 mm/s. Three ambient noise sources of 5–15 Hz, 15–35 Hz and 35–45 Hz were identified at the site and evidence suggested that these originate from pedestrian activity and vehicles on residential access and transportation roads, respectively. The evidence for this includes that more than half of the events longer than 1 s and with an amplitude above 0.05 mm/s occur at relative azimuths greater than 76°, or closer to the EW axis. Additionally, the dependence of bandwidth on event frequency also provides evidence to this point. By applying this method to gather preliminary characteristics of a site’s urban ambient seismic noise, there are potential gains in data quality by optimizing seismograph location according to the dominant sources and reduced costs through effective instrument selection. Future work should focus on optimizing the minimum acquisition time for site characterization. The integration of multiple ambient seismic data sets to reduce uncertainty regarding source location and frequencies would improve the ability of the method to identify events of interest and their respective sources.

## Figures and Tables

**Figure 1 sensors-23-02446-f001:**
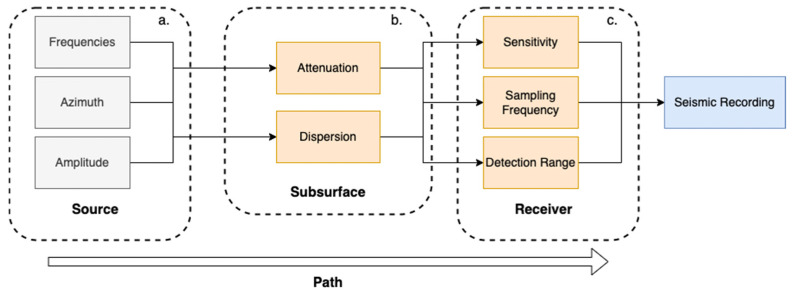
Schematic of the factors (orange) affecting the original signal produced by the source (gray). (**a**) Seismic source and its characteristics determined by the source mechanism (frequencies, azimuth, and amplitude). (**b**) Subsurface materials along the path result in attenuation and dispersion. (**c**) Receiver acquisition parameters (sensitivity, sampling frequency, and detection range).

**Figure 2 sensors-23-02446-f002:**
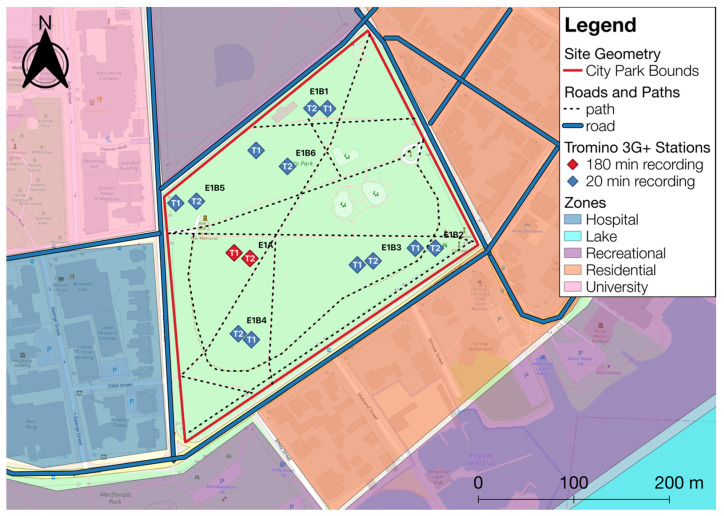
Map of City Park in Kingston, Ontario, Canada with the two seismograph locations (T1 and T2, reference locations in red), roads (solid blue lines), and pedestrian paths (dotted black lines). Colored usage zones refer to the type of activities present at the site.

**Figure 3 sensors-23-02446-f003:**
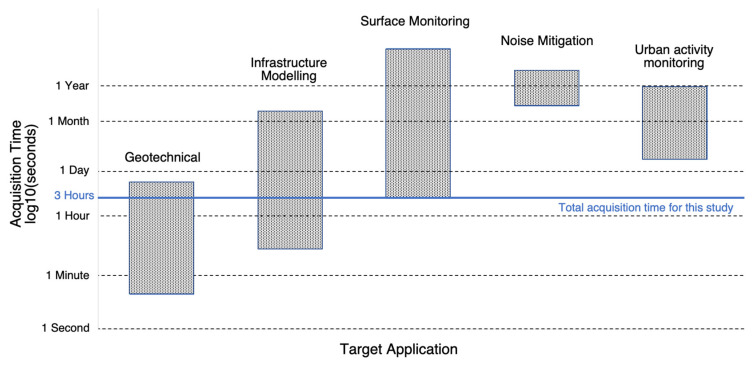
Range of acquisition time (vertical) by target application (horizontal) based on an analysis of 25 published case studies. The top and bottom of each rectangle refer to the maximum and minimum acquisition times, respectively. The vertical scale is in log_10_ (seconds). The blue line is the total recording time (3 h) used to characterize ambient seismic noise in this study, meaning that it is applicable to every target application.

**Figure 4 sensors-23-02446-f004:**
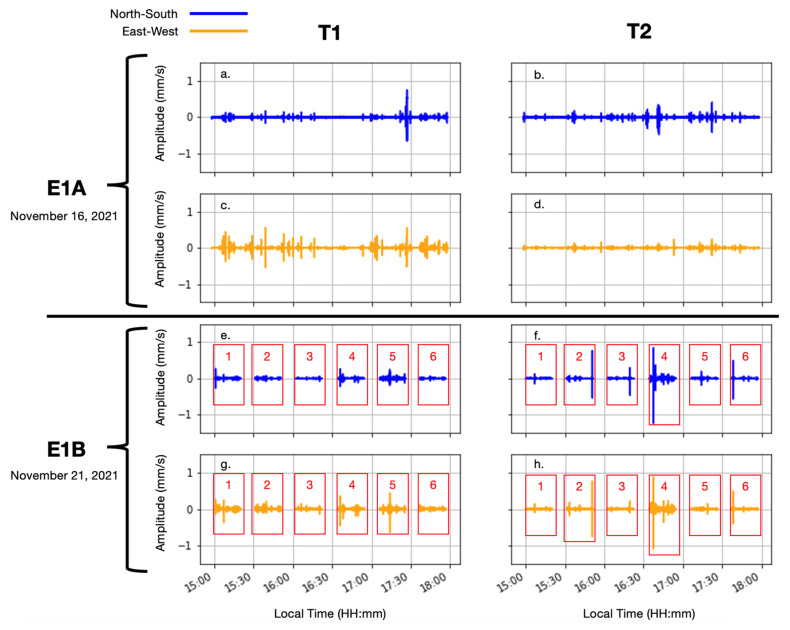
Velocity (mm/s) time series for two 3-h data collection campaigns-E1A (16 November 2021) and E1B (21 November 2021). Six different locations covered by the seismographs (T1 and T2) that were moved throughout City Park.

**Figure 5 sensors-23-02446-f005:**
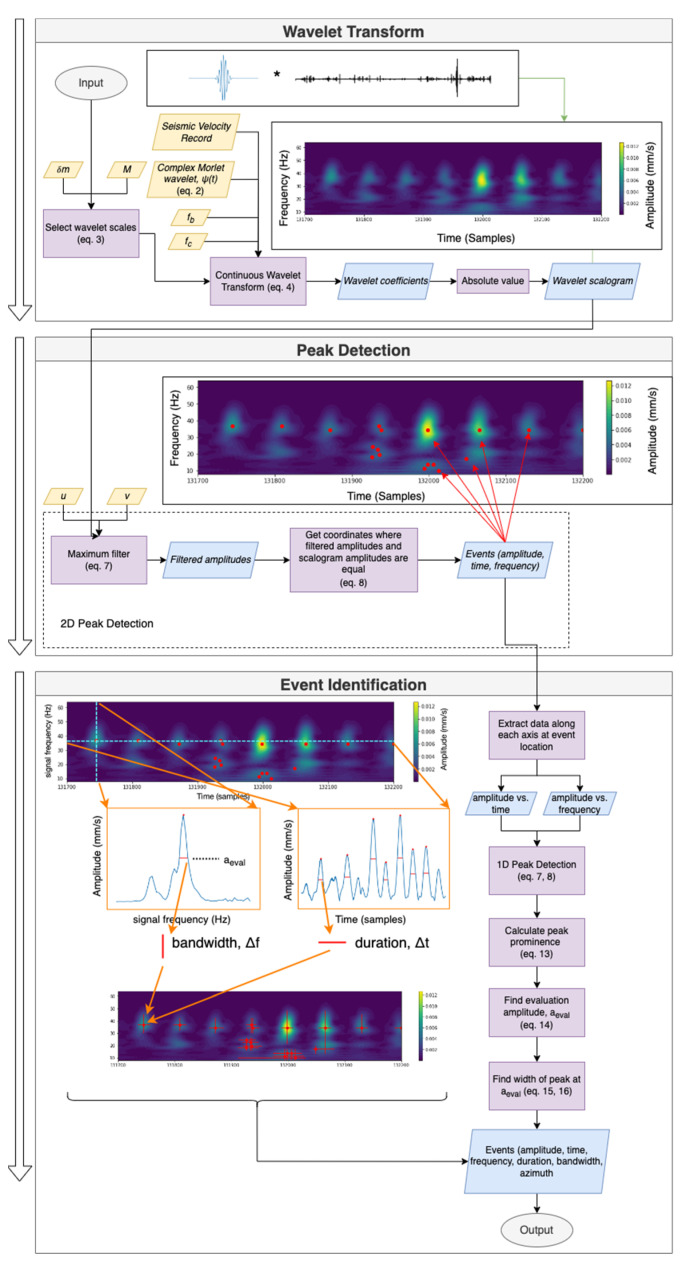
Workflow diagram for the 3-step processing methodology (wavelet transform, peak detection, and event identification) used to identify events.

**Figure 6 sensors-23-02446-f006:**
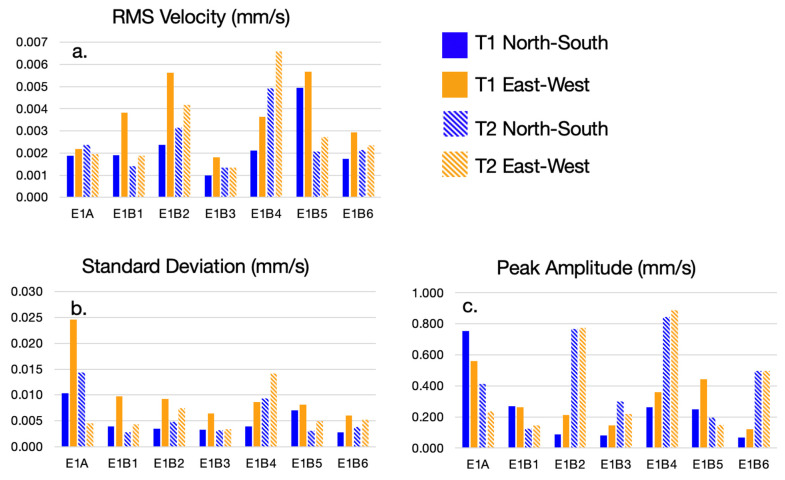
(**a**) Root mean squared velocity (mm/s), (**b**) standard deviation (mm/s), and (**c**) peak amplitude (mm/s) of all seismic data sets. The color of the bar refers to either North-South (blue) or East-West measurement (orange) and the shade refers to the seismograph station T1 (solid) or T2 (hashed).

**Figure 7 sensors-23-02446-f007:**
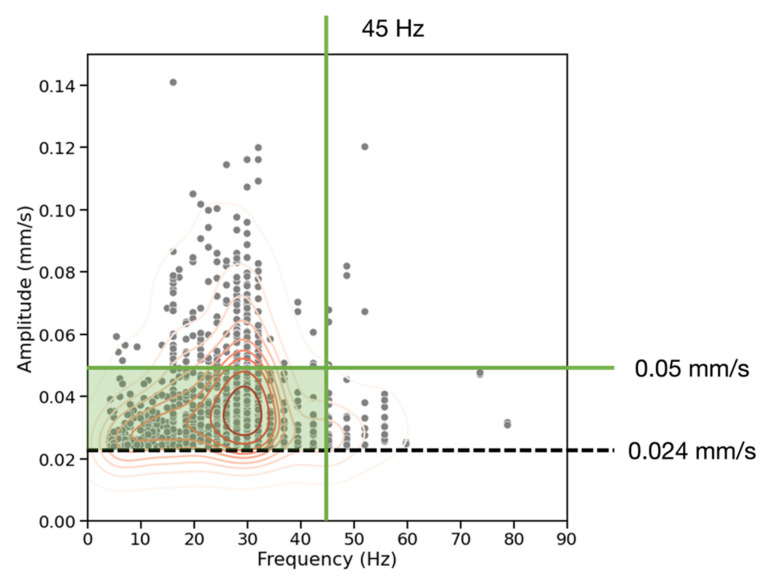
Frequency (Hz) vs. amplitude (mm/s) for all seismic records. The red contours represent the bivariate distribution of the events (gray) in steps of 10% probability density. 95% of events at the site occur at a frequency of less than 45 Hz (vertical green line). 85% of events occur at an amplitude of less than 0.05 mm/s (horizontal green line). The dashed black line represents the experimentally-selected amplitude threshold of 0.024 mm/s.

**Figure 8 sensors-23-02446-f008:**
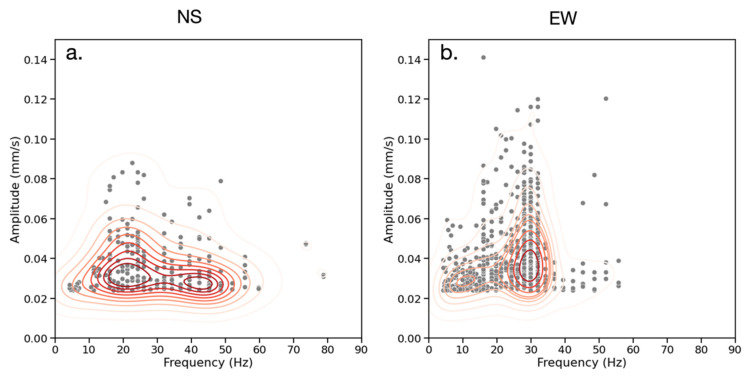
Frequency (Hz) vs. amplitude (mm/s) for all seismic records in the NS (**a**) and EW (**b**) directions. Gray dots represent events and red contours represent the bivariate distribution of events. The NS direction has a broader distribution of frequencies and lower amplitudes than the EW direction.

**Figure 9 sensors-23-02446-f009:**
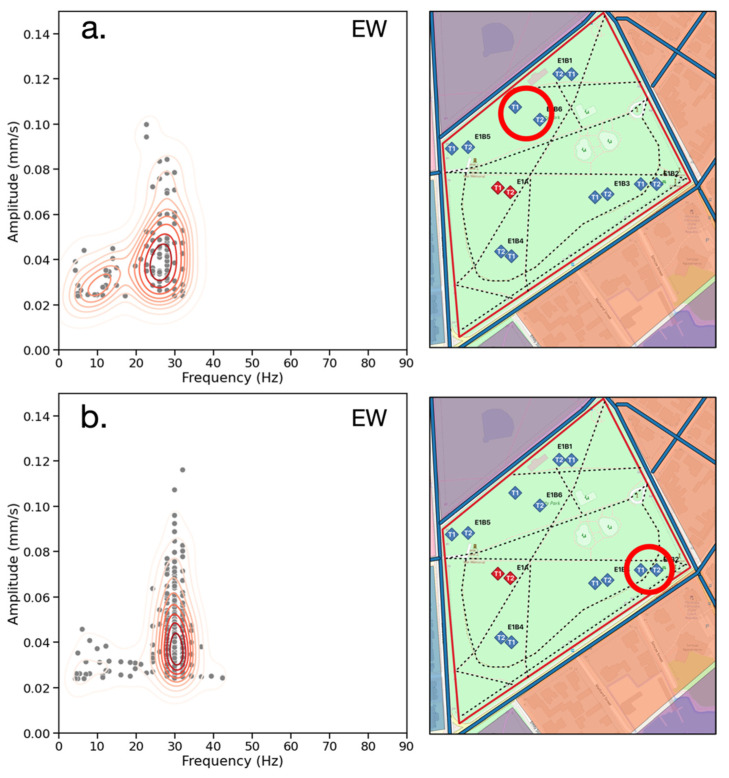
Frequency (Hz) vs. amplitude (mm/s) distributions for the EW direction for stations T1 E1B6 in the north (**a**) and E1B2 in the SE (**b**) corners of the park. These distributions highlight the spatial dependence of the frequency-amplitude distributions. The northern edge of the park exhibits two frequency clusters, at 5–15 Hz and 25–35 Hz. Higher amplitudes cluster at 25–35 Hz in the SE corner of the park.

**Figure 10 sensors-23-02446-f010:**
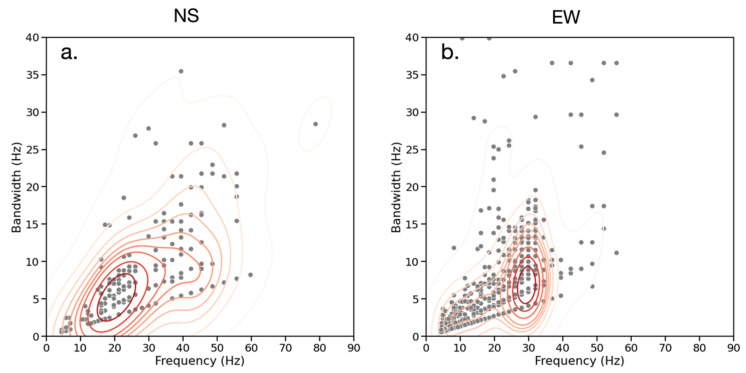
Frequency (Hz) vs. bandwidth (Hz) for all seismic records in the NS (**a**) and EW (**b**) directions. Gray dots represent events and red contours represent the bivariate distribution of events. Event bandwidth tends to increase at higher frequencies and there are two distinct clusters occurring at different bandwidths and frequencies.

**Figure 11 sensors-23-02446-f011:**
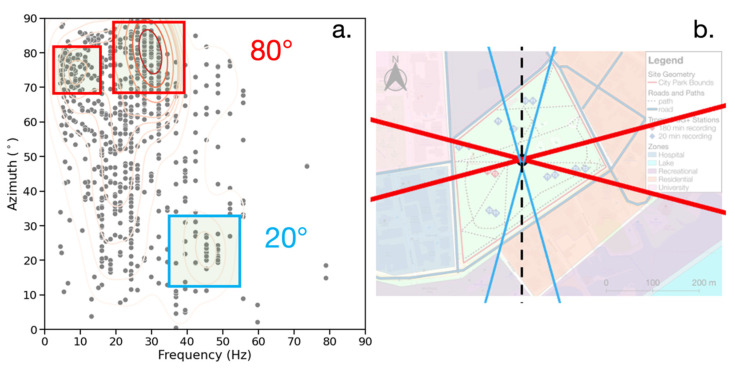
(**a**) Frequency (Hz) vs. azimuth (°) for all seismic records. Gray dots represent events and red contours represent the bivariate distribution of events. Two clusters are present at steep angles (red) and one at shallow angles (blue). (**b**) Possible source directions color-coded by angle (red is 70–80° and blue is 20°) and superimposed on the site.

**Figure 12 sensors-23-02446-f012:**
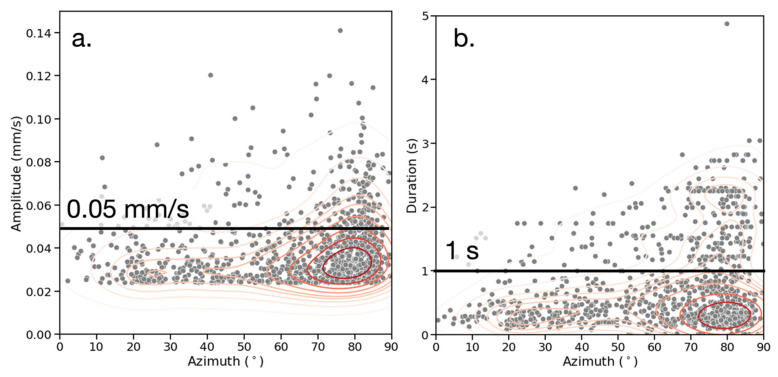
(**a**) Azimuth (°) vs. amplitude (mm/s). (**b**) Azimuth (°) vs. duration (s). Gray dots represent events, red contours represent bivariate distribution.

**Figure 13 sensors-23-02446-f013:**
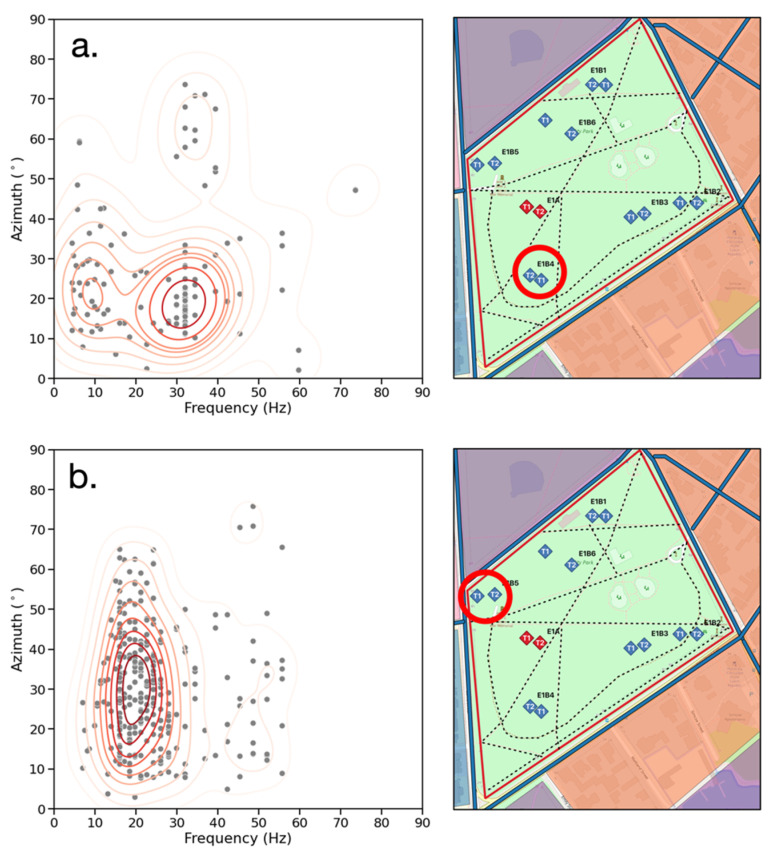
Frequency (Hz) vs. azimuth (°) distributions for individual seismograph recordings T1 E1B4 (**a**) and E1B5 (**b**). Gray dots represent events, and the red contours represent their bivariate distribution.

**Table 1 sensors-23-02446-t001:** Tromino 3G+ Seismograph manufacturer technical specifications.

Parameter	Value
Resolution	24 bits
Sensitivity	51 mV
Dynamic Range	±1.2 mm/s
Sensor Noise	0.023 mm/s at 512 Hz
Axes	XYZ
Bandwidth	0.1–1024 Hz
Temperature Range	−10 to 70 °C
Storage	4 GB

**Table 2 sensors-23-02446-t002:** Data acquisitions for each seismograph, spatially referenced to E1A T1. Each acquisition includes NS and EW axis recordings. The IDs and instruments correspond to the map in Figure 2.

ID	Instrument	Start Time (EST)	End Time (EST)	Duration (min)	Angle to Reference (°)	Distance to Reference (m)
E1A	T1	14:58:01	17:58:00	180	--	--
E1A	T2	14:58:17	17:58:16	180	104	20
E1B1	T1	15:00:13	15:20:12	20	41	180
E1B1	T2	15:00:03	15:20:02	20	36	172
E1B2	T1	15:31:01	15:51:00	20	88	190
E1B2	T2	15:31:17	15:51:16	20	88	211
E1B3	T1	16:02:07	16:22:06	20	93	130
E1B3	T2	16:02:24	16:22:23	20	92	146
E1B4	T1	16:34:33	16:54:32	20	163	90
E1B4	T2	16:34:23	16:54:22	20	174	83
E1B5	T1	17:06:19	17:26:18	20	312	80
E1B5	T2	17:06:35	17:26:34	20	313	68
E1B6	T1	17:36:58	17:56:57	20	15	110
E1B6	T2	17:37:12	17:57:11	20	39	107

**Table 3 sensors-23-02446-t003:** Experimental parameters used for event identification in City Park, Kingston.

Parameter	Value
$\mathrm{initial}\;\mathrm{scale}\;(s_{0})$	2
scale interval (δm)	0.1
number of scales (M)	60
$\mathrm{bandwidth}\;\mathrm{parameter}\;(f_{b})$	10
$\mathrm{wavelet}\;\mathrm{center}\;\mathrm{frequency}\;(f_{c})$	1
time distance interval (u)	32 samples
frequency distance interval (v)	1 sample
$\mathrm{amplitude}\;\mathrm{threshold}\;(a_{max})$	0.024 mm/s

**Table 4 sensors-23-02446-t004:** Three ambient noise sources identified based on frequency (Hz), amplitude (mm/s), azimuth (°), and bandwidth (Hz) of events at City Park, Kingston, Ontario.

Source	Frequency (Hz)	Amplitude (mm/s)	Azimuth (°)	Bandwidth (Hz)
Pedestrian	5–15	<0.03	70–80 (EW)	<5
Vehicle (Transportation)	15–35	>0.05	75–85 (EW)	5–15
Vehicle (Residential)	35–45	<0.05	15–25 (NS)	>5

## Data Availability

Seismic data available upon request.

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
