# Peer review of "Characterizing Ambient Seismic Noise in an Urban Park Environment"

_sensors, 2023, doi:10.3390/s23052446_

Round 1

Reviewer 1 Report

This paper presents a study of characterizing ambient seismic noise in an urban park environment. The paper is well written and organized. The results can be used to provide design parameters before a seismic survey. Here I only have one comment: the noise in the park environment is mainly from vehicle traffic, and the frequency is in a higher band. It should be analyzed that how to use these particular "signals" for near-surface imaging and monitoring. 

Reference: Mi, B., Xia, J., Tian, G., Shi, Z., Xing, H., Chang, X., Xi, C., Liu, Y., Ning, L., Dai, T., Pang, J., Chen, X., Zhou, C., Zhang, H., 2022. Near-surface imaging from traffic-induced surface waves with dense linear arrays: An application in the urban area of Hangzhou, China. GEOPHYSICS 87, B145–B158. https://doi.org/10.1190/geo2021-0184.1

Author Response

Done. The introduction (Section 1 in the manuscript) has been updated to include the above reference. Near-surface imaging using traffic is now listed as a distinct application alongside the others in the manuscript.

Reviewer 2 Report

The paper entitled "Characterizing ambient seismic noise in an urban park environment"is very interesting and brings novelty in the field of ambient seismic noise characterization with applicability in geotechnical studies, urban surface monitoring, modelling the seismic response of infrastructure, noise mitigation, and urban activity monitoring, which may exploit the use of well-distributed seismograph stations within an area of interest.

The developed methodology for characterizing ambient seismic events ( by parameters amplitude, frequency, time, source azimuth relative to the seismograph, duration, and bandwidths) is  original and uses a peak detection algorithm applied to the continuous wavelet transform derived amplitudes to locate the events in time and frequency space and data collected with a two-seismograph setup.

This study is well organized and the results concluded.

Author Response

Thank you for your comments.

Reviewer 3 Report

Saadia and Fotopoulos paper presents a method for evaluating ambient seismic noise characteristics in an urban park using seismographs. The workflow developed by the authors presents the basic information needed to monitor seismic response and should be useful as a guideline for conducting more detailed observations. I think it will be acceptable after only minor revisions referring to the following comments.

This study uses a pair of seismographs to conduct the observations. However, the method developed in this paper can be analyzed at a single station. Comparison of records from two simultaneous observations appears to be used only to check for amplitude variations. On the other hand, the time series of waveforms shown in Figure 4 indicates that the timing of amplitude change is considerably different, just 20 m away. Information on the time series and scalograms of the two adjacent stations may provide a more quantitative discussion of the source with the observed dominant frequencies. The advantages of observations with a pair of seismographs should be more clearly indicated in the text.

Section 2.1: Since this paper utilizes amplitude and frequency information measured by seismographs, the specifications of the Tromino3G+ (dynamic range, sensitivity, operating range, etc.) should be described in more detail.

Section 4.1: I would like to briefly describe the wave radiation characteristics due to vehicular traffic and pedestrian activity.

Line 426: I could not find the cluster corresponding to the “50 Hz signals at 25°” in Figure 13. Please check the values.
